# New Loss Functions for Fast Maximum Inner Product Search

## Abstract

Quantization based methods are popular for solving large scale maximum inner product search problems. However, in most traditional quantization works, the objective is to minimize the reconstruction error for datapoints to be searched. In this work, we focus directly on minimizing error in inner product approximation and derive a new class of quantization loss functions. One key aspect of the new loss functions is that we weight the error term based on the value of the inner product, giving more importance to pairs of queries and datapoints whose inner products are high. We provide theoretical grounding to the new quantization loss function, which is simple, intuitive and able to work with a variety of quantization techniques, including binary quantization and product quantization. We conduct experiments on public benchmarking datasets http://ann-benchmarks.com to demonstrate that our method using the new objective outperforms other state-of-the-art methods. We are committed to release our source code.

## 1 Introduction

Maximum inner product search (MIPS) has become a popular paradigm for solving large scale classification and retrieval tasks. For example, in recommendation systems, user queries and documents are embedded into dense vector space of the same dimensionality and MIPS is used to find the most relevant documents given a user query (Cremonesi et al., 2010). Similarly, in extreme classification tasks (Dean et al., 2013), MIPS is used to predict the class label when a large number of classes, often on the order of millions or even billions are involved. Lately, MIPS has also been applied to training tasks such as scalable gradient computation in large output spaces (Yen et al., 2018), efficient sampling for speeding up softmax computation (Mussmann and Ermon, 2016) and sparse updates in end-to-end trainable memory systems (Pritzel et al., 2017).

To formally define Maximum Inner Product Search (MIPS) problem, consider a database $X = \{x_i\}_{i=1,2,...,N}$ with $N$ datapoints, where each datapoint $x_i \in \mathbb{R}^d$ in a $d$-dimensional vector space. In the MIPS setup, given a query $q \in \mathbb{R}^d$, we would like to find the datapoint $x \in X$ that has the highest inner product with $q$, i.e., we would like to identify

$$x_i^* := \arg\max_{x_i \in X} \langle q, x_i \rangle.$$

Exhaustively computing the exact inner product between $q$ and $N$ datapoints is often very expensive and sometimes infeasible. Several techniques have been proposed in the literature based on hashing and quantization to solve the approximate maximum inner product search problem efficiently, and the quantization based techniques have shown strong performance (Ge et al., 2014; Babenko and Lempitsky, 2014; Johnson et al., 2017). Quantizing each datapoint $x_i$ to $\tilde{x}_i$ not only reduces storage costs and memory bandwidth bottlenecks, but also permits efficient computation of distances. It avoids memory bandwidth intensive floating point operations through Hamming distance computation and look up table operations (Norouzi et al., 2014; Jegou et al., 2011; Wu et al., 2017). In most traditional quantization works, the objective in the quantization procedures is to minimize the reconstruction error for the datapoints to be searched.

In this paper, we propose a new class of loss functions in quantization to improve the performance of MIPS. Our contribution is threefold:

- We derive a novel class of loss functions for quantization, which departs from regular reconstruction loss by weighting each pair of $q$ and $x$ based on its inner product value. We prove that such weighting leads to an effective loss function, which can be used by a wide class of quantization algorithms.

- We devise algorithms for learning the codebook, as well as quantizing new datapoints, using the new loss functions. In particular, we give details for two families of quantization algorithms, product quantization and binary quantization.

- We show that on large scale standard benchmark datasets, such as Glove100, the change of objective yields a significant gain on the approximation of true inner product, as well as the retrieval performance.

This paper is organized as follows. We first briefly review previous literature on quantization for Maximum Inner Product Search, as well as its links to $\ell_2$ nearest neighbor search in Section 2. Next, we give our main result, which is the derivation of our objective in Section 3. Applications of the new loss functions to binary quantization and product quantization are given in Section 4. Finally, we present the experimental results in Section 5.

## 2 RELATED WORKS

There is a large body of similarity search literature on inner product and nearest neighbor search. We refer readers to (Wang et al., 2014; 2016) for a comprehensive survey. Some methods also transform MIPS problem into its equivalent form of $\ell_2$ nearest neighbor using transformation such as (Shrivastava and Li, 2014; Neyshabur and Srebro, 2014), but in general are less successful than the ones that directly work in the original space. In general, these bodies of works can be divided into two families: (1) representing the data as quantized codes so that similarity computation becomes more efficient (2) pruning the dataset during the search so that only a subset of data points is considered.

Typical works in the first family include binary quantization (or binary hashing) techniques (Indyk and Motwani, 1998; Shrivastava and Li, 2014) and product quantization techniques (Jegou et al., 2011; Guo et al., 2016), although other families such as additive quantization (Babenko and Lempitsky, 2014; Martinez et al., 2016; 2018) and trenary quantization Zhu et al. (2016) also apply. There are many subsequent papers that extend these base approaches to more sophisticated codebook learning strategies, such as (He et al., 2013; Erin Liong et al., 2015; Dai et al., 2017) for binary quantization and Zhang et al. (2014); Wu et al. (2017) for product quantization. There are also lines of work that focus on learning transformations before quantization (Gong et al., 2013; Ge et al., 2014). Different from these methods which essentially minimize reconstruction error of the database points, we argue in Section 3 that reconstruction loss is suboptimal in the MIPS context, and any quantization method can potentially benefit from our proposed objective.

The second family includes non-exhaustive search techniques such as tree search (Muja and Lowe, 2014; Dasgupta and Freund, 2008), graph search (Malkov and Yashunin, 2016; Harwood and Drummond, 2016), or hash bucketing (Andoni et al., 2015) in nearest neighbor search literature. There also exist variants of these for MIPS problem (Ram and Gray, 2012; Shrivastava and Li, 2014). Some of these approaches lead to larger memory requirement, or random access patterns due to the cost of constructing index structures in addition to storing original vectors. Thus they are usually used in combination with linear search quantization methods, in ways similar to inverted index (Jegou et al., 2011; Babenko and Lempitsky, 2012; Matsui et al., 2015).

In addition, many researchers have devoted to high quality implementation of such libraries, including SPTAG Chen et al. (2018), FAISS Johnson et al. (2017), hnswlib Malkov and Yashunin (2016) etc.. We compared with ones available on `ann-benchmarks` in Section. 5.

## 3 PROBLEM FORMULATION

Common quantization techniques focus on minimizing the reconstruction error (sum of squared error) when $x$ is quantized to $\tilde{x}$. It can be shown that minimizing the reconstruction errors is equivalent to minimizing the expected inner product quantization error under a mild condition on the query distribution. Indeed, consider the quantization objective of minimizing the expected total inner product quantization errors over the query distribution:

$$\mathbb{E}_q \sum_{i=1}^{N} \|\langle q, x_i \rangle - \langle q, \tilde{x}_i \rangle\|^2 = \mathbb{E}_q \sum_{i=1}^{N} \|\langle q, x_i - \tilde{x}_i \rangle\|^2. \tag{1}$$

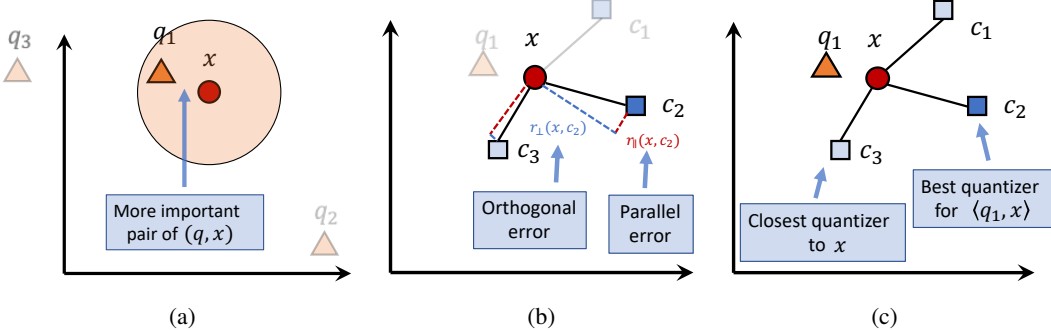

Figure 1: (a) Not all pairs of $q$ and $x$ are equally important: for $x$, it is more important to accurately quantize the inner product of $\langle q_1, x \rangle$ than $\langle q_2, x \rangle$ or $\langle q_3, x \rangle$, because $\langle q_1, x \rangle$ has a higher inner product and thus is more likely to be the maximum; (b) Quantization error of $x$ given one of its quantizer $c_2$ can be decomposed to a parallel component $r_\parallel$ and an orthogonal component $r_\perp$. Notice that $c_3$ incur more parallel loss ($r_\parallel$), while $c_2$ incur more orthogonal loss ($r_\perp$). (c) Graphical illustration of the intuition behind Equation (7). Even if $c_3$ is closer to $x$ in terms of Euclidean distance, $c_2$ is a better quantizer than $c_3$ in terms of inner product approximation error of $\langle q_1, x - c \rangle$.

Under the assumption that $q$ is isotropic, i.e., $\mathbb{E}[qq^T] = cI$, where $I$ is the identity matrix and $c \in \mathbb{R}^+$, the objective function becomes

$$\sum_{i=1}^{N} \mathbb{E}_q \|\langle q, x_i - \tilde{x}_i \rangle\|^2 = \sum_{i=1}^{N} \mathbb{E}_q (x_i - \tilde{x}_i)^T qq^T (x_i - \tilde{x}_i) = c \sum_{i=1}^{N} \|x_i - \tilde{x}_i\|^2$$

Therefore, the objective becomes minimizing the reconstruction errors of the database points $\sum_{i=1}^{N} \|x_i - \tilde{x}_i\|^2$, and this has been considered extensively in the literature.

One key observation about the above objective function (1) is that it takes expectation over all possible combinations of datapoints $x$ and queries $q$. However, it is easy to see that not all pairs of $(x, q)$ are equally important. The approximation error on the pairs which have a high inner product is far more important since they are likely to be among the top ranked pairs and can greatly affect the search result, while for the the pairs whose inner product is low the approximation error matters much less. In other words, for a given datapoint $x$, we should quantize it with a bigger focus on its error with those queries which have high inner product with $x$.

Following this key observation, we propose a new loss function by weighting the approximation error of the inner product based on the value of true inner product. More precisely, let $w(t) \geq 0$ be a monotonically non-decreasing function, and consider the following inner-product weighted quantization error

$$\sum_{i=1}^{N} \mathbb{E}_q [w(\langle q, x_i \rangle) \langle q, x_i - \tilde{x}_i \rangle^2] = \sum_{i=1}^{N} \int w(t) \mathbb{E}_q [\langle q, x_i - \tilde{x}_i \rangle^2 | \langle q, x_i \rangle = t] dP(\langle q, x_i \rangle \leq t). \quad (2)$$

One common choice can be $w(t) = \mathbf{I}(t \geq T)$, in which case we care about all pairs whose inner product is greater or equal to certain threshold $T$, and disregard the rest of the pairs.

### 3.1 NEW LOSS FUNCTION

We decompose the inner-product weighted quantization errors based on the direction of the datapoints. We show that the new loss function (2) can be expressed as a weighted sum of the parallel and orthogonal components of the residual errors with respect to the raw datapoints. Formally, let $r(x, \tilde{x}) := x - \tilde{x}$ denote the quantization residual function. Given the datapoint $x$ with $\|x\| > 0$ and its quantizer $\tilde{x}$, we can decompose the residual error into two parts, the one parallel to $x$ and the one orthogonal to $x$:

$$r_\parallel(x, \tilde{x}) := \langle x - \tilde{x}, x \rangle \cdot \frac{x}{\|x\|}, \quad (3)$$

$$r_\perp(x, \tilde{x}) := (x - \tilde{x}) - r_\parallel(x, \tilde{x}), \quad (4)$$

It's easy to see that $r(x, \tilde{x}) = r_\parallel(x, \tilde{x}) + r_\perp(x, \tilde{x})$. Next we develop our new loss function. Without loss of generality, we assume:

1. $\|q\| = 1$. The norm of $q$ does not matter to the ranking result;

2. $\|x\| \leq b$. The norm of $x$ is finite and bounded.

**Theorem 3.1.** *Assuming the query $q$ is uniformly distributed in d-dimensional unit sphere. Given the datapoint $x$ and its quantizer $\tilde{x}$, conditioned on the inner product $\langle q, x \rangle = t$ for some $t > 0$, we have*

$$\mathbb{E}_q[\langle q, x - \tilde{x}\rangle^2 | \langle q, x \rangle = t] = \frac{t^2}{\|x\|^2} ||r_\|(x, \tilde{x})||^2 + \frac{1 - \frac{t^2}{\|x\|^2}}{d - 1} ||r_\perp(x, \tilde{x})||^2. \tag{5}$$

*Proof.* First, we can decompose $q := q_\| + q_\perp$ with $q_\| := \langle q, x \rangle \cdot \frac{x}{\|x\|}$ and $q_\perp := q - q_\|$ where $q_\|$ is parallel to $x$ and $q_\perp$ is orthogonal to $x$. Then, we have

$$\begin{aligned}
\mathbb{E}_q[\langle q, x - \tilde{x}\rangle^2 | \langle q, x \rangle = t] &= \mathbb{E}_q[\langle q_\| + q_\perp, r_\|(x, \tilde{x}) + r_\perp(x, \tilde{x})\rangle^2 | \langle q, x \rangle = t] \\
&= \mathbb{E}_q[(\langle q_\|, r_\|(x, \tilde{x})\rangle + \langle q_\perp, r_\perp(x, \tilde{x})\rangle)^2 | \langle q, x \rangle = t] \\
&= \mathbb{E}_q[\langle q_\|, r_\|(x, \tilde{x})\rangle^2 | \langle q, x \rangle = t] + \mathbb{E}_q[\langle q_\perp, r_\perp(x, \tilde{x})\rangle^2 | \langle q, x \rangle = t], \quad (6)
\end{aligned}$$

The last step uses the fact that $\mathbb{E}_q[\langle q_\|, r_\|(x, \tilde{x})\rangle \langle q_\perp, r_\perp(x, \tilde{x})\rangle | \langle q, x \rangle = t] = 0$ due to symmetry. The first term of (6), $\mathbb{E}_q[\langle q_\|, r_\|(x, \tilde{x})\rangle^2 | \langle q, x \rangle = t] = ||r_\|(x, \tilde{x})||^2 \mathbb{E}_q[\|q_\|\|^2 | \langle q, x \rangle = t] = \frac{\|r_\|\|^2 t^2}{\|x\|^2}$. For the second term, since $q_\perp$ is uniformly distributed in the $(d-1)$ dimensional subspace orthogonal to $x$ with the norm $\sqrt{1 - \frac{t^2}{\|x\|^2}}$, we have $\mathbb{E}_q[\langle q_\perp, r_\perp(x, \tilde{x})\rangle^2 | \langle q, x \rangle = t] = \frac{1 - \frac{t^2}{\|x\|^2}}{d-1} ||r_\perp(x, \tilde{x})||^2$. Therefore,

$$\mathbb{E}_q[\langle q, r(x, \tilde{x})\rangle^2 | \langle q, x \rangle = t] = \frac{t^2}{\|x\|^2} ||r_\|(x, \tilde{x})||^2 + \frac{1 - \frac{t^2}{\|x\|^2}}{d - 1} ||r_\perp(x, \tilde{x})||^2. \qquad \square$$

Now we compute the inner-product weighted quantization error (2) for the case $w(t) = \mathbf{I}(t \geq T)$. One can do similar derivations for any reasonable $w(t)$. Given $T > 0$ and a datapoint $x$:

$$\mathbb{E}_q[\mathbf{I}(\langle q, x \rangle \geq T)\langle q, x - \tilde{x}\rangle^2] = f(T, \|x\|)||r_\|(x, \tilde{x})||^2 + \frac{g(T, \|x\|)}{d - 1} ||r_\perp(x, \tilde{x})||^2. \tag{7}$$

where $f(T, \|x\|)$ and $g(T, \|x\|)$ are defined as

$$f(T, \|x\|) := \int_{t=min(T, \|x\|)}^{\|x\|} \frac{t^2}{\|x\|^2} dP(\langle q, x \rangle \leq t),$$

$$g(T, \|x\|) := \int_{t=min(T, \|x\|)}^{\|x\|} (1 - \frac{t^2}{\|x\|^2}) dP(\langle q, x \rangle \leq t).$$

It is easy to see that $f(T, \|x\|)$ and $g(T, \|x\|)$ are decreasing functions of $T$ and increasing functions of $\|x\|$.

$$\begin{aligned}
\sum_{i=1}^N \mathbb{E}_q[\mathbf{I}(\langle q, x_i \rangle \geq T)\langle q, x_i - \tilde{x}_i\rangle^2] &= \sum_{i=1}^N f(T, \|x_i\|)||r_\|(x_i, \tilde{x}_i)||^2 + \frac{g(T, \|x_i\|)}{d - 1} ||r_\perp(x_i, \tilde{x}_i)||^2. \\
&\leq \sum_{i=1}^N f(T, b)||r_\|(x_i, \tilde{x}_i)||^2 + \frac{g(T, b)}{d - 1} ||r_\perp(x_i, \tilde{x}_i)||^2 \\
&\propto (d - 1)\lambda(T, b) \sum_{i=1}^N ||r_\|(x_i, \tilde{x}_i)||^2 + \sum_{i=1}^N ||r_\perp(x_i, \tilde{x}_i)||^2 \quad (8)
\end{aligned}$$

where $\lambda(T, b) = \frac{f(T, b)}{g(T, b)}$.

## 3.2 COMPUTING $\lambda(T, b)$

The new loss functions (2) we proposed is upper bounded by (8) and the equality is achieved when all of $\|x_i\| = b$ for all $x_i$. (8) can be viewed as the weighted sum of the parallel quantization errors and the orthogonal quantization errors, with respect to the original data points. Note that when $\lambda(T, b) = 1$, (8) reduces to the traditional reconstruction loss.

We can also characterize the asymptotic behavior of $\lambda(T, b)$ and show that (1) $\lambda(T, b)$ can be analytically computed, and (2) $\lambda(T, b) \to \frac{(T/b)^2}{1-(T/b)^2}$ as the dimension $d \to \infty$. In practice, we can choose $\lambda(T, b)$ empirically or through cross-validation. However, we found that $\lambda(T, b)$ when $d \to \infty$ offers a very good estimate. We discuss the sensitivity $\lambda(T, b)$ in Section. 7.5 of the Appendix.

**Theorem 3.2.** *For $b > 0, T < b$, we have $\lambda(T, b) = \frac{\int_0^{\arccos(T/b)} \sin^{d-2} \theta d\theta}{\int_0^{\arccos(T/b)} \sin^d \theta d\theta} - 1$.*

*Proof.* See the Section. 7.1 of Appendix. □

We furthermore prove that the limit of $\lambda(T, b)$ exists and that it equals $\frac{(d-1)(T/b)^2}{1-(T/b)^2}$ as $d \to \infty$. In Figure 2a, we plot $\lambda$ with $(T/b) = 0.2$ and we can see it approaches its limit quickly as $d$ grows.

**Theorem 3.3.** *When $T \geq 0$, we have $\lim_{d\to\infty} \lambda(T, b) = \frac{(T/b)^2}{1-(T/b)^2}$.*

*Proof.* See the Section. 7.2 of Appendix. □

## 4 APPLICATION TO QUANTIZATION TECHNIQUES

In this section, we derive algorithms for applying new loss functions in (8) to common quantization techniques, including vector quantization, product quantization. Discussion on binary quantization can be found in Section 7.8 of Appendix.

### 4.1 VECTOR QUANTIZATION

Recall that in vector quantization, given a set of $N$ datapoints, we want to find a codebook of size $k$ and quantize each datapoint as one of the $k$ codes. The goal is to minimize the total squared quantization error. Formally, the traditional vector quantization solves

$$\min_{\substack{c_1,c_2,...,c_k \in \mathbb{R}^d \\ \tilde{x}_i \in \{c_1,c_2,...,c_k\}, 1 \leq i \leq N}} \sum_{i=1}^{N} \|x_i - \tilde{x}_i\|^2,$$

One of the most popular quantization algorithms is the $k$-Means algorithm, where we iteratively partition the datapoints into $k$ quantizers where the centroid of each partition is set to be the mean of the datapoints assigned in the partition.

Motivated by minimizing the inner product quantization error for cases when the inner product between queries and datapoints is high, our proposed objective solves:

$$\min_{\substack{c_1,c_2,...,c_k \in \mathbb{R}^d \\ \tilde{x}_i \in \{c_1,c_2,...,c_k\}, 1 \leq i \leq N}} \sum_{i=1}^{N} \left( \mu \|r_\parallel(x_i, \tilde{x}_i)\|^2 + \|r_\perp(x_i, \tilde{x}_i)\|^2 \right), \tag{9}$$

where $\mu = (d - 1)\lambda(T, b)$ is a hyperparameter as a function of $d$ and $T$ following (8).

We solve (9) through a $k$-Means style Lloyd's algorithm, which iteratively minimizes the new loss functions by assigning datapoints to partitions and updating the partition quantizer in each iteration. The assignment step is computed by enumerating each quantizer and finding the quantizer that minimizes (9). The update step finds the new quantizer $\tilde{x}^* \in \mathbb{R}^d$ for a partition of datapoints $x_1, x_2, \ldots, x_m \in \mathbb{R}^d$, i.e.,

$$\tilde{x}^* = \min_{\tilde{x} \in \mathbb{R}^d} \sum_{i=1}^{m} \left( \mu \|r_\parallel(x_i, \tilde{x})\|^2 + \|r_\perp(x_i, \tilde{x})\|^2 \right), \tag{10}$$

Because of the changed objective, the best quantizer is no longer the center of the partition. Since (10) is a convex function of $\tilde{x}$, there exists an optimal solution. The update rule given a fixed partitioning can be found by setting the partial derivative of (10) with respect to each codebook entry to zero. This algorithm provably converges in a finite number of steps. See Algorithm 1 in Appendix for a complete outline of the algorithm. Note that, in the special case that $\mu = 1$, it reduces to regular $k$-Means algorithm.

**Theorem 4.1.** *The optimal solution of* (10) *is*

$$\tilde{x}^* := \mu \left( \mathbb{1} + \frac{\mu - 1}{m} \sum_{i=1}^{m} \frac{x_i x_i^T}{\|x_i\|^2} \right)^{-1} \frac{\sum_{i=1}^{m} x_i}{m}. \tag{11}$$

*Proof.* See Section. 7.3 of the Appendix. $\square$

**Theorem 4.2.** *Algorithm 1 converges in finite number of steps.*

*Proof.* This immediately follows from the fact that the loss defined in (9) is always non-increasing during both assignment and averaging steps under the changed objective. $\square$

## 4.2 PRODUCT QUANTIZATION

A natural extension of vector quantization is product quantization, which works better in high dimensional spaces. In product quantization, the original vector space $\in \mathbb{R}^d$ is decomposed as the Cartesian product of $m$ distinct subspaces of dimension $\frac{d}{m}$, and vector quantizations are applied in each subspace separately [1]. For example, let $x \in \mathbb{R}^d$ be written as

$$x = (x^{(1)}, x^{(2)}, \ldots, x^{(m)}) \in \mathbb{R}^d,$$

where $x^{(j)} \in \mathbb{R}^{\frac{d}{m}}$ is denoted as the sub-vector for the $j$-th subspace. We can quantize each of the $x^{(j)}$ to $\tilde{x}^{(j)}$ with its vector quantizer in subspace $j$, for $1 \le j \le m$. With product quantization, $x$ is quantized as $(\tilde{x}^{(1)}, \ldots, \tilde{x}^{(m)}) \in \mathbb{R}^d$ and can be represented compactly using the assigned codes.

Using our proposed loss objective (8), we minimize the following loss function instead of the usual objective of reconstruction error:

$$\min_{\substack{C_1, C_2, \ldots, C_m, \\ A \in \{1,2,\ldots,k\}^{N \times m}}} \sum_{i=1}^{N} \left( \mu \|r_\|(x_i, \tilde{x}_i)\|^2 + \|r_\perp(x_i, \tilde{x}_i)\|^2 \right), \tag{12}$$

where $\tilde{x}_i$ denotes the product quantization of $x_i$, i.e.,

$$\tilde{x}_i := (C_{1, A_{i,1}}, C_{2, A_{i,2}}, \ldots, C_{m, A_{i,m}}).$$

To optimize (12), we apply the vector quantization of Section 4.1 over all subspaces, except that the subspace assignment is chosen to minimize the global objective over all subspaces (12), instead of using the objective in each subspace independently. Similarly, the update rule is found by setting the derivative of loss in (12) with respect to each codebook entry to zero. The complete algorithm box of Algorithm (1) is found in Section 7.6 of the Appendix.

## 5 EXPERIMENTS

In this section, we show our proposed quantization objective leads to improved performance on maximum inner product search. First, we compare using productization mechanism with reconstruction loss and our proposed loss to show that the new loss leads to better retrieval performance and more accurate estimation of maximum inner product values. Secondly, we compare in fixed-bit-rate settings with `QUIPS`, which achieves state-of-the-art in multiple MIPS tasks. Finally, we analyze the end-to-end MIPS retrieval performance of our algorithm in terms of speed-recall trade-off in controlled hardware environment and timing. We follow the benchmark setting from `ann-benchmarks`, which provides 11 competitive baselines with pre-tuned parameters. We plot benchmarks speed-recall curve and show our algorithm achieves the state-of-the-art.

---

[1]Random rotation or permutation of the original vectors can be done before doing the Cartisean product.

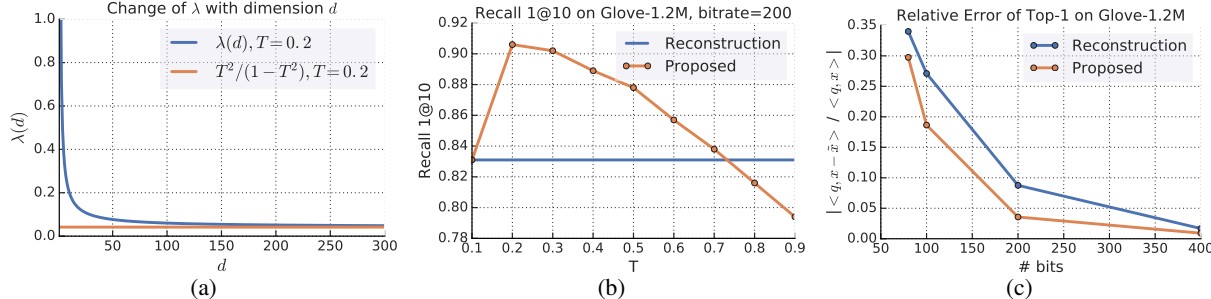

Figure 2: (a) $\lambda(T = 0.2, b = 1)$ in Theorem 3.2 computed analytically as function of $d$ using recursion of (13), quickly approaches its limit. Therefore we use its limit for computing $\lambda$ in our experiments. (b) The retrieval Recall1@10 for different $T$ when $b = 1$. (c) The relative error of inner product estimation for true Top-1 on `Glove1.2M` dataset, across multiple number of bits settings.

### 5.1 ANALYSIS OF THE PROPOSED LOSS FUNCTION

We compare the result of same quantization mechanism with different loss functions (reconstruction loss and the proposed loss (8)). We use `Glove1.2M` which is a collection of 1.2 million 100-dimensional word embeddings trained with the method described in Pennington et al. (2014), and we provide rationale on using `Glove1.2M` for evaluation in Section 7.9 of the Appendix. We set $\lambda$ as $\lim_{d\to\infty} \lambda(T = 0.2, b = 1)$ for all our experiments.

Figure. 2b illustrates the Recall1@10 of product quantization on `Glove1.2M`, with reconstruction loss and proposed loss. We can see that our algorithm leads to large improvement over the one with reconstruction loss. In addition to retrieval, many application scenarios also require estimating the value of the inner product $\langle q, x \rangle$. For example, in softmax functions, inner product values are often logits and are later used to compute the probability. One direct consequence of (8) is that the objective weighs pairs by their importance and thus leads to lower estimation error on top-ranking pairs. We measure $|\frac{\langle q,x \rangle - \langle q,\tilde{x} \rangle}{\langle q,x \rangle}|$ as the relative error on true inner product. New objective clearly produces smaller relative error over all bitrate settings (Figure. 2c).

### 5.2 MAXIMUM INNER PRODUCT SEARCH RETRIEVAL

Next, we show our MIPS retrieval performance with fixed number of bits. We compare to that of `QUIPS` Guo et al. (2016) which achieves the state-of-the-art on MIPS tasks. `QUIPS` describes three variants, `QUIPS-Cov(x)`, `QUIPS-Cov(q)` and `QUIPS-Opt` which uses covariance of database vectors, covariance of query vectors and sample queries respectively. In Figure 3. We measure the performance at fixed bitrate by Recall 1@N, which corresponds to the fraction that Top-1 ground truth result is recalled in $N$ retrieved results. Clearly the results using our proposed loss function out-performs all of the variants in `QUIPS`.

Other quantization methods may benefit from new loss function by switching from reconstruction. For example, binary quantization such as Dai et al. (2017) uses reconstruction loss in its original paper, which can be easily swapped for the proposed loss by one line change of the loss objective. We show the results which illustrated the improvement of loss function in Section 7.8 of Appendix. It is possible that other quantization methods also see a moderate improvement. We will discuss our attempt with Local Search Quantization (LSQ) Martinez et al. (2018) in Section 7.7 of the Appendix.

### 5.3 EXTREME CLASSIFICATION INFERENCE

Extreme classification with large number of classes requires evaluating the last layer (classification layer) with all possible classes. When there are $\mathcal{O}(M)$ classes, this becomes a major computation bottleneck as it involves huge matrix multiplication followed by Top-K. Thus this is often solved using Maximum Inner Product Search to speed up the inference. We evaluate our methods on extreme classification using the `Amazon-670k` dataset Bhatia et al. (2015). An MLP classifier is trained over 670,091 classes, where the last layer has a dimensionality of 1,024. We evaluate retrieval performance on the classification layer and show the results in Figure. 3c, by comparing it against brute force matrix multiplication.

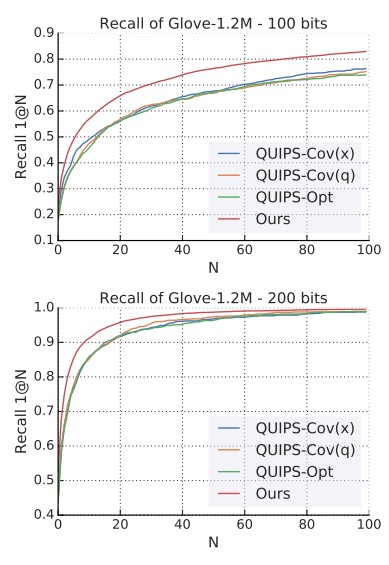

(a) MIPS recall on `Glove1.2M`.

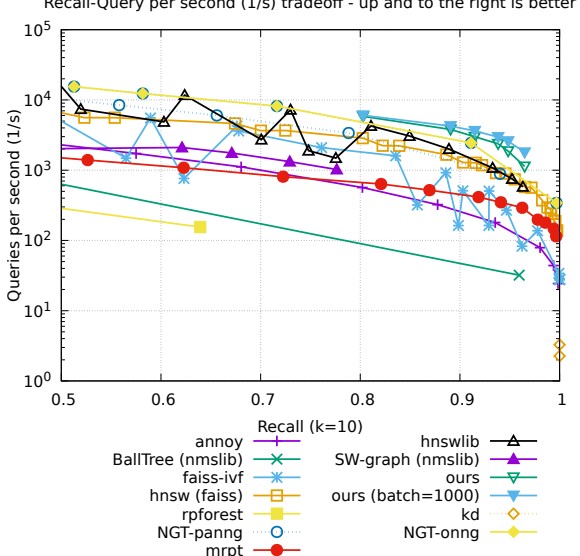

(c) Speed-recall trade-off on `Glove1.2M` Recall@10

| Bitrate | 1@1 | 1@10 | 1@100 | Bitrate | 1@1 | 1@10 | 1@100 |
|---------|------|-------|--------|---------|------|-------|--------|
| 256 bits, PQ | 0.652 | 0.995 | 0.999 | 512 bits, PQ | 0.737 | 0.998 | 1.000 |
| 256 bits, Ours | **0.656** | **0.996** | **1.000** | 512 bits, Ours | **0.744** | 0.997 | 1.000 |
| 1024 bits, PQ | 0.778 | 1.000 | 1.000 | 2048 bits, PQ | 0.782 | 1.000 | 1.000 |
| 1024 bits, Ours | **0.812** | 1.000 | 1.000 | 2048 bits, Ours | **0.875** | 1.000 | 1.000 |

(b) Extreme classification on `Amazon670k`.

Figure 3: (a) Recall 1@N curve on `Glove1.2M` comparing with variants of `QUIPS` Guo et al. (2016) on MIPS tasks. (b) Extreme classification on `Amazon670k` performed through MIPS, compared to baseline product quantization that uses reconstruction loss. (c) Recall-Speed benchmark with 11 baselines from Aumüller et al. (2019) on `Glove1.2M`. The parameters of each baselines are pre-tuned and released on: `http://ann-benchmarks.com/`. Our approach compare favorably on speed-recall trade-off over popular state-of-the-art, production ready methods.

## 5.4 RECALL-SPEED BENCHMARK

Fixed-bit-rate experiments mostly compare asymptotic behavior, and often overlook preprocessing overhead such as learned rotation or lookup table computation, which can be substantial. To evaluate effectiveness of MIPS algorithms in realistic setting, it is important to perform end-to-end benchmarks and compare the speed-recall curve. We adopted the methodology of public benchmark `ANN-benchmarks` Aumüller et al. (2019), which plots a comprehensive set of 11 algorithms for comparison, including `faiss` Johnson et al. (2017) and `hnswlib` Malkov and Yashunin (2016).

Our benchmarks are conducted on same platform of Intel Xeon W-2135 with one CPU single thread, and followed the benchmark's protocol. Our implementation builds on product quantization with the proposed quantization and SIMD based ADC Guo et al. (2016) for distance computation. This is further combined with a vector quantization based tree Wu et al. (2017), and our curve is plotted by varying the number of leaves to search in the tree. Figure 3c shows our performance on `Glove1.2M` significantly outperforms the competing methods, especially in high recall region, where Recall of 10 is over 80%. We are committed our open source our implementation and parameter tunings.

## 6 CONCLUSION

In this paper, we propose a new quantization loss function for inner product search, which replaces traditional reconstruction error. The new loss function is weighted based on the inner product values, giving more weight to the pairs of query and database points with higher inner product values. The proposed loss function is theoretically proven and can be applied to a wide range of quantization methods, for example product and binary quantization. Our experiments show superior performance on retrieval recall and inner product value estimation, compared to methods that use reconstruction error. The speed-recall benchmark on public datasets further indicates that the proposed method outperform state-of-arts baselines which are known to be hard to beat.

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

# 7 APPENDIX

## 7.1 PROOF OF THEOREM 3.2

*Proof of Theorem 3.2.* Let $\theta := \arccos t$, and $\alpha := \arccos(T/b)$. Note that $\frac{dP(\langle q,x \rangle \le t)}{dt}$ is proportional to the surface area of $(d-1)$-dimensional hypersphere with a radius of $\sin \theta$. Thus we have $\frac{dP(\langle q,x \rangle = t)}{dt} \propto S_{d-1} \sin^{d-2} \theta$, where $S_{d-1}$ is the surface area of $(d-1)$-sphere with unit radius.

Thus, $\lambda(T, b)$ can be re-written as:

$$\lambda(T, b) = \frac{\int_0^\alpha \cos^2 \theta S_{d-1} \sin^{d-2} \theta d\theta}{\int_0^\alpha \sin^2 \theta S_{d-1} \sin^{d-2} \theta d\theta} = \frac{\int_0^\alpha \sin^{d-2} \theta d\theta}{\int_0^\alpha \sin^d \theta d\theta} - 1.$$

Denote $I_d = \int_0^\alpha \sin^d \theta d\theta$,

$$I_d = -\cos \alpha \sin^{d-1} \alpha + \int_0^\alpha \cos^2 \theta (d-1) \sin^{d-2} \theta d\theta$$

$$= -\cos \alpha \sin^{d-1} \alpha + (d-1) \int_0^\alpha \sin^{d-2} \theta d\theta - (d-1) \int_0^\alpha \sin^d \theta d\theta$$

$$= -\cos \alpha \sin^{d-1} \alpha + (d-1) I_{d-2} - (d-1) I_d.$$

This gives us a recursive formula to compute $I_d$ when $d$ is a positive integer :

$$I_d = \frac{-\cos \alpha \sin^{d-1} \alpha}{d} + \frac{d-1}{d} I_{d-2} \tag{13}$$

With the base case of $I_0 = \alpha$, and $I_1 = 1 - \cos \alpha$, the exact value of $\lambda = \frac{I_{d-2}}{I_d} - 1$ can be computed explicitly in $\mathcal{O}(d)$ time. $\square$

## 7.2 PROOF OF THEOREM 3.3

*Proof of Theorem 3.3.* First, it is easy to see that because $\sin^{d-2} \alpha < \sin^d \alpha$, $\frac{I_{d-2}}{I_d} < 1$. Next, from Cauchy–Schwarz inequality for integrals, we have

$$\left( \int_0^\alpha \sin^{d+2} x \sin^{d-2} x dx \right)^2 \le \int_0^\alpha \sin^{d+2} x dx^2 \int_0^\alpha \sin^{d-2} x dx^2$$

Rearranging this we have $\frac{I_d}{I_{d+2}} \le \frac{I_{d-2}}{I_d}$, which proves that $\frac{I_{d-2}}{I_d}$ is monotonically non-increasing. Given that it has a lower bound and is monotonically non-increasing, the limit of $\frac{I_d}{I_{d+2}}$ exists.

Dividing both sides of Equation. 13 by $I_d$, we have

$$1 = \frac{-\cos \alpha \sin^{d-1} \alpha}{d I_d} + \frac{(d-1) I_{d-2}}{d I_d}$$

Thus $\lim_{d \to \infty} \frac{(d-1) I_{d-2}}{d I_d} = \lim_{d \to \infty} \frac{I_{d-2}}{I_d} > 1$ exists. And therefore $\lim_{d \to \infty} \frac{\cos \alpha \sin^{d-1} \alpha}{d I_d} > 0$ also exists. Furthermore,

$$\lim_{d \to \infty} \frac{\frac{\cos \alpha \sin^{d-1} \alpha}{d I_d}}{\frac{\cos \alpha \sin^{d-3} \alpha}{(d-2) I_{d-2}}} = 1 \Rightarrow \lim_{d \to \infty} \frac{(d-2) I_{d-2}}{d I_d} = \frac{1}{\sin^2 \alpha}$$

Finally we have $\lim_{d \to \infty} \lambda = \frac{1}{\sin^2 \alpha} - 1 = \frac{T/b}{1 - (T/b)^2}$, and this proves Theorem 3.3. $\square$

## 7.3 PROOF OF THEOREM 4.1

*Proof of Theorem 4.1.* Indeed,

$$
\begin{aligned}
g(x_i, \tilde{x}) &:= \mu \|r_\|(x_i, \tilde{x})\|^2 + \|r_\perp(x_i, \tilde{x})\|^2 \\
&= \mu \frac{((\tilde{x} - x_i)^T \cdot x_i)^2}{\|x_i\|^2} + \|(\tilde{x} - x_i) - \frac{(\tilde{x} - x_i)^T x_i}{\|x_i\|} \frac{x_i}{\|x_i\|}\|^2 \\
&= (\tilde{x} - x_i)^T (\tilde{x} - x_i) + (\mu - 1) \frac{((\tilde{x} - x_i)^T x_i)^2}{\|x_i\|^2} \\
&= (\tilde{x}^T \tilde{x} - 2 x_i^T \tilde{x} + x_i^T x_i) + (\mu - 1) \frac{(\tilde{x}^T x_i - x_i^T x_i)^2}{\|x_i\|^2} \\
&= (\tilde{x}^T \tilde{x} - 2 x_i^T \tilde{x} + x_i^T x_i) + (\mu - 1) \frac{(\tilde{x}^T x_i)^2 + \|x_i\|^4 - 2 \tilde{x}^T x_i \|x_i\|^2}{\|x_i\|^2} \\
&= (\tilde{x}^T \tilde{x} - 2 x_i^T \tilde{x} + x_i^T x_i) + (\mu - 1) \left( \frac{(\tilde{x}^T x_i)^2}{\|x_i\|^2} + \|x_i\|^2 - 2 \tilde{x}^T x_i \right) \\
&= \|\tilde{x}\|^2 + (\mu - 1) \frac{(\tilde{x}^T x_i)^2}{\|x_i\|^2} - 2 \mu \tilde{x}^T x_i + \mu \|x_i\|^2 \\
&= \tilde{x}^T \tilde{x} + (\mu - 1) \frac{\tilde{x}^T x_i x_i^T \tilde{x}}{\|x_i\|^2} - 2 \mu \tilde{x}^T x_i + \mu \|x_i\|^2.
\end{aligned}
$$

Therefore, the derivative of $g(x_i, \tilde{x})$ with respect to $\tilde{x}$ is

$$
\frac{dg(x_i, \tilde{x})}{d\tilde{x}} = 2\tilde{x} + 2 \frac{\mu - 1}{\|x_i\|^2} x_i x_i^T \tilde{x} - 2 \mu x_i.
$$

Setting $\sum_{i=1}^m \frac{dg(x_i, \tilde{x})}{d\tilde{x}} = 0$, we have

$$
\sum_{i=1}^m \frac{dg(x_i, \tilde{x})}{d\tilde{x}} = 0
$$

$$
\iff \sum_{i=1}^m \left( 2\tilde{x} + 2 \frac{\mu - 1}{\|x_i\|^2} x_i x_i^T \tilde{x} - 2 \mu x_i \right) = 0
$$

$$
\iff \left( \mathbb{1} + \frac{\mu - 1}{m} \sum_{i=1}^m \frac{x_i x_i^T}{\|x_i\|^2} \right) \tilde{x} = \mu \frac{\sum_{i=1}^m x_i}{m}
$$

$$
\iff \tilde{x} = \mu \left( \mathbb{1} + \frac{\mu - 1}{m} \sum_{i=1}^m \frac{x_i x_i^T}{\|x_i\|^2} \right)^{-1} \frac{\sum_{i=1}^m x_i}{m}.
$$

$\square$

## 7.4 ALGORITHM FOR VECTOR QUANTIZATION WITH THE MODIFIED OBJECTIVE

---

**Algorithm 1** Proposed Vector Quantization Algorithm For Minimizing Weighted Quantization Errors

---

**Input:**

- A set of $N$ datapoints $x_1, x_2, \ldots, x_N \in R^d$.
- A scalar $\mu > 0$, the weight for quantization error components tradeoff (selection guided by desired inner product threshold $T$ in Sec 3).
- A positive integer $k$, the size of the codebook.

**Output:**

- A set of codebook $c_1, c_2, \ldots, c_k \in \mathbb{R}^d$
- Partition assignment $a_1, a_2, \ldots, a_N \in \{1, 2, \ldots, k\}$ for the $N$ datapoints such that $x_i$ is quantized as $c_{a_i}$.

**Algorithm:**
Initialize the centroids $c_1, c_2, \ldots, c_k$ by choosing $k$ random datapoints.
Set new_error $= +\infty$.
**do**

    old_error $\leftarrow$ new_error.
    [**Partition Assignment**]
    **for each** $i \in \{1, 2, \ldots, N\}$ **do**

$$a_i \leftarrow \arg\min_j \left( \mu \|r_\|(x_i, c_j)\|^2 + \|r_\perp(x_i, c_j)\|^2 \right)$$

    **end for**
    [**Centroid Update**]
    **for each** $i \in \{1, 2, \ldots, k\}$ **do**

$$S \leftarrow \{x_j | a_j = i, 1 \le j \le N\}$$

$$c_i \leftarrow \mu \left( \mathbb{1} + \frac{\mu - 1}{|S|} \sum_{x \in S} \frac{xx^T}{\|x\|^2} \right)^{-1} \frac{\sum_{x \in S} x}{|S|},$$

    where $|S|$ denote the cardinality of the set $S$.
    **end for**
    Compute new_error $\leftarrow \sum_{i=1}^{N} \left( \|r_\|(x_i, c_{a_i})\|^2 + \lambda \|r_\perp(x_i, c_{a_i})\|^2 \right)$
**while** old_error $<$ new_error
Output the codebook $c_1, c_2, \ldots, c_k$ and the assignment $a_1, a_2, \ldots, a_N$.

---

## 7.5 SENSITIVITY TO CHOICE OF $T$

Another interesting question is how the retrieval performance relates to $T$, given $\lambda(T, b)$. Intuitively, if $T$ is set too low, then the objective takes almost all pairs of query and database points into consideration, and becomes similar to the standard reconstruction loss. If $T$ is too high, then very few pairs will be considered and the quantization may be inaccurate for low value inner product pairs. Figure 2b shows the retrieval evaluation of `Glove1.2M` dataset under different $T$. We use $T = 0.2$ for all of our retrieval experiments.

### 7.6 Codebook Optimization in Product Quantization

For example, consider the first vector $C_{1,1}$ in the codebook $C_1$ for the first subspace, and let $x_i$ be one of the datapoints the first subspace of which is encoded as $C_{1,1}$, i.e., $x_i^{\widetilde{(1)}} = C_{1,1}$. Write $x_i$ as $x_i = (x_i^{(1)}, x_i^{(-1)})$, and $\tilde{x}_i$ as $\tilde{x}_i = (C_{1,1}, x_i^{\widetilde{(-1)}})$. Then

$$\mu\|r_\parallel(x_i, \tilde{x}_i)\|^2 + \|r_\perp(x_i, \tilde{x}_i)\|^2 = \tilde{x}_i^T \tilde{x}_i + (\mu - 1)\frac{\tilde{x}_i^T x_i x_i^T \tilde{x}_i}{\|x_i\|^2} - 2\mu \tilde{x}_i^T x_i + \mu\|x_i\|^2$$

$$= (C_{1,1}^T C_{1,1} + x_i^{\widetilde{(-1)}^T} x_i^{\widetilde{(-1)}}) - 2\mu(C_{1,1}^T x_i^{(1)} + x_i^{\widetilde{(-1)}^T} x_i^{(-1)}) + \mu\|x_i\|^2$$

$$+ (\mu - 1)\frac{C_{1,1}^T x_i^{(1)} x_i^{(1)^T} C_{1,1} + 2x_i^{\widetilde{(-1)}^T} x_i^{(-1)} x_i^{(1)^T} C_{1,1} + x_i^{\widetilde{(-1)}^T} x_i^{(-1)} x_i^{(-1)^T} x_i^{\widetilde{(-1)}}}{\|x_i\|^2}.$$

Let $S$ denote the set of indices $S := \{i | A_{i,1} = 1\}$. Therefore, the partial derivative of (12) with respect to $C_{1,1}$ is

$$\sum_{i \in S} \left( 2C_{1,1} + (\mu - 1)\frac{2x_i^{(1)} x_i^{(1)^T} C_{1,1} + 2x_i^{(1)} x_i^{(-1)^T} x_i^{\widetilde{(-1)}}}{\|x_i\|^2} \right). \tag{14}$$

Set (14) to be zero, and we have

$$C_{1,1}^* = \mu \left( \mathbb{1} + \frac{\mu - 1}{|S|} \sum_{i \in S} \frac{x_i^{(1)} x_i^{(1)^T}}{\|x_i\|^2} \right)^{-1} \frac{\sum_{i \in S}(x_i^{(1)} - \frac{(1 - \frac{1}{\mu})x_i^{(1)} x_i^{(-1)^T} x_i^{\widetilde{(-1)}}}{\|x_i\|^2})}{|S|}. \tag{15}$$

### 7.7 Notes on applying new loss function on LSQ

Local Search Quantization (LSQ) has shown strong results on achieving state-of-the-art performance on `SIFT1M` with fixed-bit-rates. Since LSQ also uses reconstruction loss, it is possible that the new loss would provide additional improvement over it. The author released the code on Github built on CUDA and Julia. However we ran into multiple issues with julia-0.7.0 and GeForce GTX 1080 (8G of GPU RAM), which seems to be related internals of Julia's CuArray library that causes OOM. Even though we tried our best to fix the issues, we weren't able to apply this on LSQ. We refer readers to the open issues of LSQ's code on `https://github.com/una-dinosauria/Rayuela.jl/issues/` for more context. We would love to update and release the results if the issues are resolved.

## 7.8 RESULTS ON BINARY QUANTIZATION

Another popular family of quantization function is binary quantization. In such a setting, a function $h(x) := \mathbb{R}^d \to \{0,1\}^h$ is learned to quantize datapoints into binary codes, which saves storage space and can speed up distance computation. There are many possible ways to design such a binary quantization function, and some Carreira-Perpinán and Raziperchikolaei (2015); Dai et al. (2017) uses reconstruction loss.

It is straight-forward to apply our proposed loss function there and the details of algorithm. We follow the setting of Stochastic Generative Hashing (SGH) Dai et al. (2017), which explicitly minimizes reconstruction loss and has been shown to outperform earlier baselines. In their paper, a binary auto-encoder is learned to quantize and dequantize binary codes:

$$\tilde{x} = g(h(x)); \text{where } h(x) \in \{0,1\}^h$$

where $h(\cdot)$ is the "encoder" part which binarizes original datapoint into binary space and $g(\cdot)$ is the "decoder" part which reconstructs the datapoints given the binary codes. The authors of the paper uses $h(x) = sign(W_h^T x + b_h)$ as the encoder function and $g(h) = W_g^T h$ as the decoder functions, respectively. The learning objective is to minimize the reconstruction error of $||x - \tilde{x}||^2$, and the weights in the encoder and decoder are optimized end-to-end using standard stochastic gradient descent. Following our discussion in Section. 3, this is an suboptimal objective in the MIPS settings, and the learning objective given in (8) should be preferred. This implies minimal changes to the algorithm of Dai et al. (2017) except the loss function part, while holding everything else unchanged. We show below the results of SGH Dai et al. (2017) and SGH with the new loss function to illustrate the proposed loss function can be easily applied to binary quantization case.

| SIFT1M | 1@1 | 1@10 | 10@10 | 10@100 |
|---|---|---|---|---|
| 64 bits, SGH | 0.028 | 0.096 | 0.053 | 0.220 |
| 64 bits, SGH-newloss | **0.071** | **0.185** | **0.093** | **0.327** |
| 128 bits, SGH | 0.073 | 0.195 | 0.105 | 0.376 |
| 128 bits, SGH-newloss | **0.196** | **0.406** | **0.209** | **0.574** |
| 256 bits, SGH | 0.142 | 0.331 | 0.172 | 0.539 |
| 256 bits, SGH-newloss | **0.362** | **0.662** | **0.363** | **0.820** |

## 7.9 Idiosyncrasy of datasets for MIPS / NNS evaluation

With the increasing interest in Maximum Inner Product Search problem (MIPS) and its practical value in embedding based retrieval systems, researchers compete intensively on evaluation for publications. The widely used datasets for evaluating these include `SIFT1M/1B`, `GIST1M`, `Glove1.2M` for NNS, as well as `Movielens` and `Netflix` for MIPS. However, we argue many datasets have data distribution issues and can lead to misinformed conclusions about the relative performance of algorithms. Some of these datasets suffer from (1) serve representation redundancy; (2) heterogeneous importance across dimensions. Since algorithms were often developed based on the empirical observation of these datasets, some have adapted to exploit the idiosyncrasies of particular datasets. Our suggestion is to evaluate embedding vectors which show less data distribution issues, and thus methods can be compared on the merit of their effectiveness against well-formed vectors, and revisit some of comparisons.

**SIFT1M, SIFT1B and GIST1M** are introduced by Jegou et al. (2011) to illustrate the use of product quantization. SIFT is a keypoint descriptor while GIST is image-level descriptor, which have been hand-crafted for image retrieval. These vectors have a high correlation between dimensions and has a high degree of redundantly. The intrinsic dimension of SIFT1M and GIST are much lower than its dimensionality.

**Movielens and Netflix** dataset are formed from the SVD of the rating matrix of Movielens and Netflix websites, respectively. This is introduced by Shrivastava and Li (2014) for MIPS retrieval evaluation. Following SVD of $X = (U\Lambda^{1/2T})(\Lambda^{1/2}V)$, the dimension of these two datasets correspond to the eigenvalues of $X$. Thus the variance of dimensions are sorted by eigenvalues, and the first few dimensions are much more important than later ones. Moreover, the datasets are $\mathcal{O}(10k)$ in size and can hardly be called large-scale for evaluation purposes.

**Glove1.2M** is a word embeddings dataset similar to word2vec, which use neural-network style training with a bottleneck layer. Similarly we also created Amazon670k which comes from a feedforward network. These datasets exhibit less data distribution problem as illustrated in In fact, it is our general observation that bottleneck layer lead to independent dimensions with similar entropy, making them good datasets for benchmarking.

Below we plot the correlation and variance by dimensions of `SIFT1M`, `GIST1M`, `MovielensSVD`, `NetflixSVD`, as well as `Glove1.2M` and `Amazon670k`. It is obvious that `SIFT1M` and `GIST1M` has strong correlations between dimensions and the intrinsic dimensions are significantly lower than its original dimension. On the hand `MovielensSVD` and `NetflixSVD` suffers from problem of variance of numeric scales across dimensions, as shown in the graph below. With huge variance in the numeric scale, a few dimensions are dominating the results and the rest of dimension play much smaller roles and can almost be discarded. Thus we argue, despite the popular of first four datasets, has strong structure in data distribution and are not effective representation. On such datasets, it is difficult to judge the performance of a MIPS / NNS algorithm, because the dataset idiosyncrasy may be exploited to provide much gain to the extend it may overwhelm the effect of searching strategies.

We argue that the role of a MIPS / NNS algorithm should be efficient search against well-formed data, not fixing representational deficiency of features in the dataset. On the other hand, dataset such as `Glove1.2M`, `Amazon670k`, or empirically many neural network embedding has less distribution structures. This is perhaps because bottleneck layers encourage each dimension to encode independent information and carry similar amount of entropy. We suggest that these datasets should be used for benchmarking, under reproducible settings such as `http://www.ann-benchmarks.`

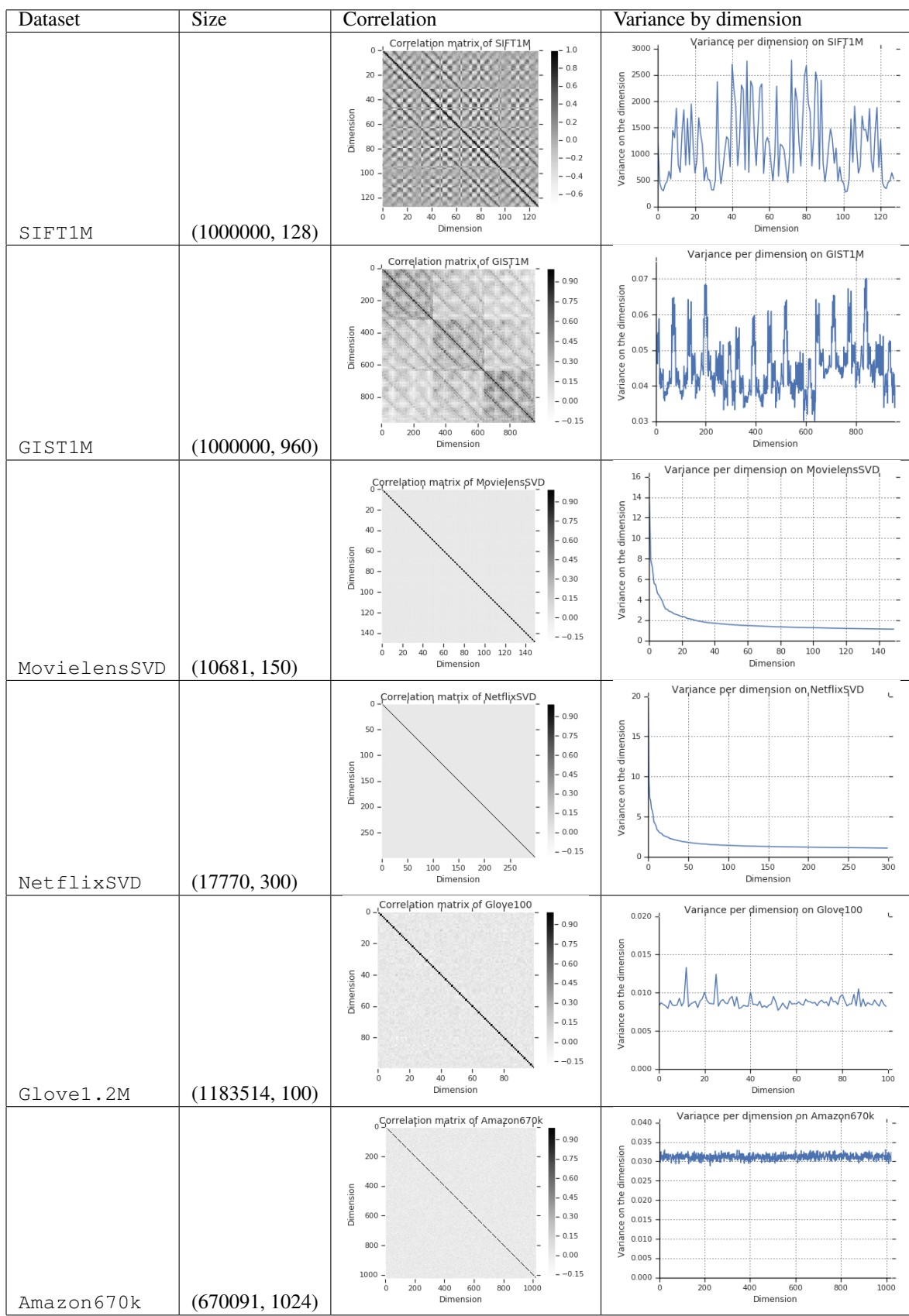

