# OpenReview forum: "New Loss Functions for Fast Maximum Inner Product Search"
_ICLR.cc/2020/Conference — Reject_

### Official Review · AnonReviewer1 · 2019-10-18
**Official Blind Review #1**

**Rating:** 3

**Review:**

Summary:
The authors propose a new loss function for solving large-scale inner product search that rely on quantization, based on the intuition that all pairs of (query, database vector) are not equally important for a given query. In particular, the authors weight the reconstruction error so that the pairs with a higher scalar product are more precisely quantized (as they lie among the most plausible candidates).

Strengths of the paper:
- The paper is well written and easy to follow. In particular, the intuition of the method is well explained in Figure 1 and the setup in Section 3 is well formulated.
- The proposed method works with a variety of quantization approaches, such as binary, simple PQ or LSQ (even if the authors aren't able to report results for this last method due to technical issues as explained in Appendix 7.7)

Weaknesses of the paper:
- The related work could be more detailed, see for example: "Spreading vectors for Similarity Search", Sablayrolles et al. ; "Pairwise Quantization",  Babenko et al ; "Unsupervised Neural Quantization for Compressed-Domain Similarity Search", Morozov et al.

Justification of rating:
The paper proposes a new loss function that weights the scalar products differently according to their importance than can be applied to a wide range of existing quantization methods. However, the strength of experimental results (in particular the fact that LSQ or other cited references above are missing) remains unclear.

**Experience Assessment:**

I do not know much about this area.

**Review Assessment: Checking Correctness Of Derivations And Theory:**

I assessed the sensibility of the derivations and theory.

**Review Assessment: Checking Correctness Of Experiments:**

I carefully checked the experiments.

**Review Assessment: Thoroughness In Paper Reading:**

I read the paper at least twice and used my best judgement in assessing the paper.

---

> ### Author Response · Authors · 2019-11-07
> **Response to Official Blind Review #1**
>
> 1) Thanks for the suggestion of related papers and we will cite those in our paper. Note that some of paper’s are on arXiv very recently.
>
> 2) It appears that the reviewer’s main concern is the lack of LSQ baseline: (1) We have benchmarked against state-of-the art MIPS algorithms such as QUIPS. (2) We conducted recall-speed experiment on widely used public ann-benchmarks which contains 12 baselines including highly competitive faiss and hsnw, and have shown significant improvement over them (pls see Figure3(c)). Thus, we feel our experiments are comprehensive and state-of-the-art.
>
> For reproduction of the results using LSQ, even though Julia GPU version did not work (as discussed in Appendix 7.7), after the deadline, we were able to use a numpy reimplementation of code. We give the comparison below on Glove1.2M:
>
> Bitrate=200       PQ         Ours      LSQ
>
> Recall1@1         0.427     0.524     0.479
> Recall1@10       0.831     0.906     0.882
> Recall1@100     0.983     0.996     0.995
>
> Our method gives higher recall than LSQ. We will incorporate the results in the final version.

---

### Official Review · AnonReviewer3 · 2019-10-21
**Official Blind Review #3**

**Rating:** 3

**Review:**

The paper proposes a novel quantization algorithm for maximum inner product search. Instead of imposing equal weights on all possible queries, the proposed framework considers giving larger weights on queries having larger inner product values. The authors shows that this derives a loss function having different weights on parallel and orthogonal component of residual of quantization. The paper shows the performance of the algorithm mainly by using the Glove1.2M dataset.

The basic idea and the weighting approach would be reasonable. Empirical evaluation might be slightly weak.

In practice, how can the hyper-parameter (T and b) be determined? T = 0.2 shows the best performance on Fig. 2 (b), but its seems sensitive, and generality of this setting is not clear.

In (7), only <q,x> larger than T is penalized, but larger <q,\tidel{x}> seems important as well. If the encoded \tilde{x} has a large inner product value wrongly, it would deteriorate performance.

The tightness of the the upper bound of the inequality in the end of page 4 would be unclear. Replacing norm with its possible maximum value seemingly has large effect.

Minor comment:
Figure 1 is not referred from the main text.

In Fig. 2 (b), reconstruction and proposed have the save value at T = 0.1. Is this just a coincident?

**Experience Assessment:**

I do not know much about this area.

**Review Assessment: Checking Correctness Of Derivations And Theory:**

I did not assess the derivations or theory.

**Review Assessment: Checking Correctness Of Experiments:**

I assessed the sensibility of the experiments.

**Review Assessment: Thoroughness In Paper Reading:**

I read the paper at least twice and used my best judgement in assessing the paper.

---

> ### Author Response · Authors · 2019-11-07
> **Response to Official Blind Review #3**
>
> Thanks for the valuable feedback! Our response are listed below:
> 1) Sensitivity of T and b.
> b is the upper bound of the datapoint norms, therefore it is not a tuneable hyperparameter but a deterministic value given the dataset. For Glove, b=1.
> For T, it is an estimate of “the expected distance of Top-L result”, where L is some value larger than K.
>
> Empirically the final result isn’t sensitive to T for a wide range of L. For example in Glove dataset, if we set L to anywhere from 100 to10000, we arrive at roughly similar value of T (around 0.2) and they all yield sizeable improvement on retrieval recall. Thus, the results are not sensitive to precise tuning of T.
>
> L=100: T=0.323, Recall1@10=0.901; L=1000: T=0.24; Recall1@10=0.902;
> L=200: T=0.301, Recall1@10=0.907; L=2000: T=0.219, Recall1@10=0.902;
> L=500: T=0.271, Recall1@10=0.905; L=5000: T= 0.179, Recall1@10=0.901;
>
> 2) Formulation did not account for error on <q, \tilde{x}>.
> The reviewer is worried that we are not penalizing <q, \tilde{x}> when <q, x> is less than T. In our formulation, we quantize x to \tilde{x} so that the pairs of (q, x) whose inner product value is larger is penalized more. In addition, one can use a smooth weighting function instead of the step function, such as w(t) = exp(-h * t). However, we did not find that to be a problem in our experiments.
>
>
> 3) Tightness of bound.
> For most neural network generated datasets, the norm of the vectors typically do not vary too much. If one is worried that it may happen, multiscale trick [Wu et al, NIPS 2017] can be used, but this is orthogonal to the contribution of this paper and highly dataset dependent.
>
> 4) Minor comments:
> 1) We will refer to Figure 1 in the main text in the final version.
> 2) Yes, the fact T=0.0 and T=0.1 has almost the same value is coincidental. The rough value of T Recall1@10 are:
> T=0.0: 0.831, T=0.1: 0.831, T=0.2: 0.906, T=0.3: 0.902, T=0.4: 0.889 …

---

### Official Review · AnonReviewer2 · 2019-10-24
**Official Blind Review #2**

**Rating:** 6

**Review:**

The paper proposes new loss functions for quantization when the task of interest is maximum inner product search (MIPS).
The paper is well written with clear descriptions, fairly comprehensive analysis and empirical exploration, and good results, and in general I agree that learning quantization so as to minimize quantization related errors on task at hand is a good strategy.
Specific comments and suggestions for strengthening the paper are:
a) The proposed loss function in (2) includes a weight function that serves as a proxy for the task objective of giving more emphasis to quantization errors on samples with larger inner product.  Instead, why not use the true task objective which for the MIPS task is stated in the Introduction section?  If this was considered please comment on reasons for not including / discussing this in the paper, otherwise perhaps this’ll be good to discuss.
b) Did the authors consider using a task dependent training data set which will capture both ‘q’ and ‘x’ distributions and potentially lead to even further improved quantization?  This has the disadvantage of making quantization dependent on query distribution, but in cases where such data is available it will be very valuable to know if incorporating data distributions in quantization process helps performance and to what extent.
c) It will also be valuable to consider the closely related task of cosine distance based retrieval and comment on how that impacts the modifications of loss functions.
d) The idea of learning quantization under objective of interest using observed data distribution has been studied earlier (e.g. see Marcheret et al., “Optimal quantization and bit allocation for compressing large discriminative feature space transforms,” ASRU 2009), perhaps worth citing as related work.


**Experience Assessment:**

I have published in this field for several years.

**Review Assessment: Checking Correctness Of Derivations And Theory:**

I assessed the sensibility of the derivations and theory.

**Review Assessment: Checking Correctness Of Experiments:**

I assessed the sensibility of the experiments.

**Review Assessment: Thoroughness In Paper Reading:**

I read the paper at least twice and used my best judgement in assessing the paper.

---

> ### Author Response · Authors · 2019-11-07
> **Response to Official Blind Review #2**
>
> Thanks the reviewer for the constructive feedback!
> a) Using ranking loss directly is considered in QUIPS-opt in Guo et al. 16 but in experiments it performed worse than the formulation in this paper. The main reason is that it focuses too much on the relative ordering of the top few and is sensitive to noise and small changes in query distribution.
> b) This is also considered in QUIPS-opt and QUIPS-cov(q), which we compared to in Figure3(a). We perform better than both across all bitrate.
> c) Under unit-norm dataset, cosine distance and dot product are equivalent. Thus our algorithm directly for Cosine similarity without modification.
> d) We will cite the provided reference. Thanks!

---

### Decision · Program_Chairs · 2019-12-19

**Decision:**

Reject

**Comment:**

While there was some support for the ideas presented, the majority of reviewers felt that this submission is not ready for publication at ICLR in its present form.

Concerns were raised as to the generality of the approach, thoroughness of experiments, and clarity of the exposition.